# Development of a Tailored, Complex Intervention for Clinical Reflection and Communication about Suspected Urinary Tract Infections in Nursing Home Residents

**DOI:** 10.3390/antibiotics9060360

**Published:** 2020-06-25

**Authors:** Sif H. Arnold, Julie A. Olesen, Jette N. Jensen, Lars Bjerrum, Anne Holm, Marius B. Kousgaard

**Affiliations:** 1The Section of General Practice and Research Unit for General Practice, Department of Public Health, University of Copenhagen, Øster Farimagsgade 5, Building 24 Q, K 1353 Copenhagen, Denmark; julie.olesen@sund.ku.dk (J.A.O.); lbjerrum@sund.ku.dk (L.B.); anneholm@sund.ku.dk (A.H.); marbro@sund.ku.dk (M.B.K.); 2Department of Clinical Microbiology, Herlev and Gentofte Hospital, University of Copenhagen, Herlev Ringvej 75, 2730 Herlev, Denmark; jette.nygaard.jensen.01@regionh.dk

**Keywords:** urinary tract infection, nursing home, antibiotic resistance, drug prescription, implementation barriers, communication barriers, primary care

## Abstract

Background: Inappropriate antibiotic treatments for urinary tract infections (UTIs) in nursing homes cause the development of resistant bacteria. Nonspecific symptoms and asymptomatic bacteriuria are drivers of overtreatment. Nursing home staff provide general practice with information about ailing residents; therefore, their knowledge and communication skills influence prescribing. This paper describes the development of a tailored, complex intervention for a cluster-randomised trial that targets the knowledge of UTI and communication skills in nursing home staff to reduce antibiotic prescriptions. Methods: A dialogue tool was drafted, drawing on participatory observations in nursing homes, interviews with stakeholders, and a survey in general practice. The tool was tailored through a five-phase process that included stakeholders. Finally, the tool and a case-based educational session were tested in a pilot study. Results: The main barriers were that complex patients were evaluated by healthcare staff with limited knowledge about disease and clinical reasoning; findings reported to general practice were insignificant and included vague descriptions; there was evidence of previous opinion bias; nonspecific symptoms were interpreted as UTI; intuitive reasoning led to the inappropriate suspicion of UTI. Conclusion: Sustainable change in antibiotic-prescribing behaviour in nursing homes requires a change in nursing home staff’s beliefs about and management of UTIs.

## 1. Introduction

For the elderly living in nursing homes in Europe, the primary reason for prescribing an antibiotic is urinary tract infection (UTI), but the scientific literature regards many of these prescriptions as inappropriate [1,2,3]. Antibiotics are indispensable drugs, but their use causes the development of resistant bacteria [4,5]. Therefore, preserving the effectiveness of antibiotics by limiting unnecessary use is a public health priority, and antibiotic stewardship is one way of achieving this goal [6].

The literature identifies two sources of antibiotic overtreatment of UTIs in nursing homes: treatment of nonspecific symptoms and asymptomatic bacteriuria [7]. Health professionals often consider nonspecific behavioural symptoms as an indication of UTIs in the elderly, but guidelines recommend that these symptoms should not be treated with antibiotics [8,9,10,11]. Asymptomatic bacteriuria is the presence of bacteria in a urine sample from a patient that shows no signs or symptoms originating from the urinary tract [12]. Positive urine tests in the nursing home population are common because the frequency of asymptomatic bacteriuria varies between 40–80% among nursing home residents [9,13]. The evidence states that this patient group will not benefit from antibiotic treatment of asymptomatic bacteriuria. Moreover, a positive test will often be misinterpreted as the patient having a UTI [12,14,15]. Consequently, indiscriminate use of urine testing of nursing home residents can lead to overtreatment.

In Danish nursing homes, the majority of the nursing home staff are healthcare helpers or healthcare assistants, who are present around the clock [16]. Nurses, employed in fewer numbers, primarily work day shifts [17]. Healthcare helpers undergo 19 months of schooling after their compulsory education, which can be shortened with additional basic schooling. Healthcare helpers become healthcare assistants by continuing their education for an additional 20 months. While learning about diseases and clinical reasoning is a central part of nursing education, it is virtually absent in the education of a healthcare helper and assistant [16]. Healthcare helpers and assistants attend to the residents’ everyday needs, and the nurse only gets involved when helpers or assistants find that a resident seems different than usual. If nursing home staff suspect a UTI to be the cause, they can contact the physician. The nursing home staff provide the physician with the clinical history, and the physician often prescribes antibiotics without seeing the patient [18,19]. Consequently, the staff’s knowledge and communication skills directly influence diagnosis and treatment [20,21]. Therefore, we developed an intervention for the nursing home staff that improve their knowledge about UTIs and refine their ability to communicate relevant clinical observations to the physician. The ultimate goal of the intervention is to decrease the incidence of misdiagnoses of UTIs and thereby reduce antibiotic prescriptions to nursing home residents.

Antibiotic stewardship programs generate mixed results, using education, decision algorithms, and communication tools to reduce inappropriate antibiotic prescribing for UTIs in nursing homes [22]. Studies have shown that uptake of antibiotic stewardship can be inadequate, perhaps because attention to implementation barriers is neglected [23,24]. “Tailored” interventions are planned interventions that follow an investigation of factors that explain current practices and seek to uncover reasons underlying the resistance to new practices [25]. Tailoring is recommended to increase the uptake and effect of interventions by adapting it to settings and users [26].

This paper will describe the development process of a tailored, complex intervention for a cluster-randomised trial and the assumptions behind the intervention. The tailoring process is aimed at adjusting the intervention to nursing home settings by identifying and addressing barriers to implementation.

## 2. Results

### 2.1. Organizational Challenges of Diagnosing UTIs in Nursing Homes

We identified three typical routes of communication about suspected UTIs in nursing home patients: telephone, email, and direct face-to-face contact. Nursing home staff have reported telephone calls as the most common and face-to-face meetings as the least common form of communication. Email communication followed the same pattern as telephone communication; hence, the model depicted in Figure 1 is expected to cover the vast majority of inquiries.

Typically, only healthcare helpers or assistants attend the patient first-hand. Therefore, they are the first to notice if a nursing home resident appears different than usual. Healthcare helpers pass their observations along to a healthcare assistant or nurse, who are the only ones allowed to contact the general practitioner (GP) directly. The only time the GP answers the phone directly is 8–9 a.m. At all other times, the medical secretary responds and delivers the message to the GP, who decides on the treatment.

The three main implications of the communication pathway relevant to the initial draft of the dialogue tool are (1) the communication pathway between resident and GP includes several different persons with various professional backgrounds. This emphasises the need for structured clinical handovers, as information is lost with each additional actor in the communication pathway [27,28]. (2) the model showed a one-way communication pathway from the patients’ bedside to the GP. If the GP lacks information about the patient, the pathway has to be reversed, which is difficult and time-consuming for all actors involved. (3) Healthcare helpers or assistants are always involved in providing the clinical history for the GP.

The implications provide a deeper insight into why overtreatment of UTIs is a persistent issue: While healthcare helpers’ and assistants’ knowledge about diseases and clinical reasoning are limited, their job requires them to evaluate highly complex patients. Thus, to bypass the nursing home staff’s judgment, the nursing homes and general practice have developed a simplistic system. This system weighs objective, measurable results of urinary tests higher than subjective assessments of signs and symptoms when making the diagnosis. However, in doing so, they perpetuate the faulty notion that asymptomatic bacteriuria is UTI; hence, this simplistic system sustains and exacerbates overtreatment.

### 2.2. From the Original Idea to the Final Intervention

The original idea included four components. We intended to combine a decision aid (based on Loeb et al.) with a communication tool (based on McMaughan et al. and Ydemann) that included the result of a C-reactive protein (CRP) point-of-care-test (POC-test) [29,30,31,32]. CRP is a test to identify severe bacterial infections. The nursing home staff would learn to apply the tool and the test in educational sessions.

During the development process, the research group decided to create a dialogue tool to collect information systematically and structure clinical communication. Accordingly, the educational component should be case-based, introduce the dialogue tool, and address knowledge gaps to improve clinical reasoning. In addition, the POC-test was discarded because the research group feared that by using it without the basic clinical thinking and communication skills being in place, this could lead to misuse, confusion, and an overcomplicated implementation process.

The final intervention included three components: a reflection tool, a communication tool, and a case-based education session (Table 1). The decision aid was changed to a reflection tool with three sections: observations of signs and symptoms, a decision aid flowchart, and discussion. The communication tool included five sections: Identification, Situation, Background, Assessment, and Recommendation (ISBAR). Collectively, the reflection and the communication tools were called “the dialogue tool”.

When the patient presents with fever and the absence of symptoms from other organ systems, the flowchart proposed by Loeb et al. does not consider urinary catheter use [29]. We changed the flowchart slightly, making it more similar to the consensus algorithm developed by van Buul et al. [33]. In general, the educational level varies within the nursing home staff group; therefore, at every stage of the development process, we edited the length of sentences and words for clarity. For example, “dysuria” was changed to “pain when urinating”.

### 2.3. The Dialogue Tool

The next section describes the early draft of the dialogue tool and the findings and adjustments from the tailoring process and the pilot.

#### 2.3.1. The Reflection Tool

##### Early Draft

In the early draft (Appendix A), we translated the diagnostic algorithm originally developed for physicians and nurses by Loeb et al. into Danish and adapted a vocabulary understandable to healthcare helpers, assistants, and nurses alike [29]. We separated observations from the decision algorithm and installed checkboxes and room for notes on vital signs. Loeb’s decision algorithm determines when to order a urine culture. In Denmark, the GP makes this decision. Therefore, we changed the conclusion of the flowchart to decide if a UTI was likely. Despite the controversial role of the urinary dipstick in diagnostics of UTI, the research group included the test result in Section 1 of the reflection tool under the subheading “Vital Signs” [34,35]. GPs request the test; hence, discouraging dipstick use could cause conflict with the GPs and lead to a decreased use of the tool. We emphasised that the urinary dipstick result was optional, not required. We added a section with “Other Observations” that contained other causes of the nonspecific symptoms, such as changes in medications. In the semi-structured interviews, some of the GPs reported that they thought the nursing home staff sometimes contacted general practice too early in the illness. Therefore, we added suggestions for actions before contacting the GP. These were increased observation of the resident and preventive hygienic measures, e.g., improved intimate hygiene.

##### Barriers to Implementation of the Reflection Tool

The Staff’s Intuitive Reasoning Led to Inappropriate Suspicion of UTIs

During the interviews with nursing home staff, we tested the dialogue tool, specifically in the clinical case of an elderly nursing home resident with nonspecific behavioural changes and smelly urine (see Text B1 in Appendix A). The participants correctly noted that urinary symptoms were absent in the observation section. However, when they used the flowchart, they consistently concluded that UTI was likely. This was surprising because the case was designed to lead to the conclusion that UTI was unlikely.

In the focus group interview, a nurse assistant described that when she realised that urinary symptoms were absent, she ignored the possible paths in the flowchart, and, misinterpreting light confusion as “one or more constitutional symptoms”, she concluded that UTI was likely (Figure 2). We understood her response to mean that her interpretation of the case had made her unable to follow the paths outlined in the flowchart. The same thing happened in phase 5 of the tailoring process, where it became clear that the nurse equated mild confusion with delirium and saw this as a symptom of UTI.

In Phase 4, the informant still concluded that UTI was likely despite realizing that urinary symptoms were absent. When the informant was confronted with the discrepancy, her reaction displayed fear of missing a UTI diagnosis; this illustrates how hard it is for healthcare professionals to disregard a positive urinary test:
“But I don’t think you can do that. I don’t think you can… because… no but this is the reason you need to be careful with this… Because they are a little confused, and then you have the lady here, who has foul smelling urine and she has nitrite and leucocytes, so you can’t exclude that there is something there or something coming, and therefore of course I would give her lots of fluids and then observe her and send a (urine sample for, red) culture and resistance, because ehm, she could develop something. Especially, when she is a little more confused than usual. You can’t exclude it you know…”

To ensure that it was not the interview guide or the design of the dialogue tool that was causing the informants to reach the wrong conclusion, a number of adjustments were made. In Phase 4, the dialogue tool was meticulously reviewed with the informant before introducing the case to make sure that unfamiliarity with the tool was not the cause. In addition, the nurse from Phase 5 was explicitly asked to abstain from using her clinical intuition and only consider the facts of the case and the questions in the flowchart. The design of the flowchart was adjusted to tie the flowchart and the observation section closer together.

The term “constitutional symptoms” seemed to trigger the suspicion of UTIs more frequently than other terms. Twice, we saw a switch of paths in the flowchart because the informants wanted to use “one or more constitutional symptoms”. Therefore, we changed this to “severe symptoms”. In several phases, the informants requested a checkbox for symptoms that were less severe than the description of delirium:
“Because I wouldn’t say that the symptoms she has are acute. So, I would like to have a space where I could state something like ‘general’ (symptoms, red).”

This indicated that the nursing home staff were working with a spectrum of nonspecific behavioural symptoms ranging from delirium to slight change. The understanding was that delirium requires immediate medical attention, and slight change requires observation and preventive measures such as ensuring fluid intake. We added a checkbox for nonspecific symptoms to accommodate the need and a definition of delirium to avoid confusion of the two.

The case (see Text B1 in Appendix A) describes the presence of smelly and unclear urine, which nursing home staff referred to as a sign of UTI. Evidence suggests, however, that in the case of these residents, smelly or unclear urine is actually not a diagnostic sign of UTI [36]. Regardless, we chose to include smelly and unclear urine in the observations section, because most informants searched for space to make a note of the information, we assumed that leaving it out could become a barrier to the use of the dialogue tool.

From Phases 4 and 5, it was clear that it could be difficult to change the informants’ intuitive understandings of what constituted a UTI. In the pilot, the head nurse noted that discussions about the residents with suspected UTIs deepened her understanding of the UTI definition. Therefore, we specified that the discussion mentioned in Section 3 of the reflection tool should include a colleague. The purpose was to make healthcare helpers, healthcare assistants, and nurses discuss their findings, check their thinking, and learn from each other in the process. If done consistently, this feature could lead to discussions that would embed the new way of approaching UTIs at the nursing homes.

Reported Symptoms Were often Known and Insignificant Changes

The first draft already highlighted that all observations should be new onsets. However, several stakeholders emphasised that the nursing home staff would often report already known, persisting symptoms and insignificant deviations from the norm. The GP said:
“Well, we often go on home visits where we think ‘why were we called? There wasn’t anything new here?’”

For emphasis, we added the word “new onset” to the headlines in all boxes of the observations section and to the relevant text boxes in the flowchart. We also added this consideration as a discussion point in the discussion section.

#### 2.3.2. The Communication Tool

##### Early Draft

The ISBAR communication model is widely used in the Danish healthcare sector. The first draft was a combination of a Danish ISBAR used in hospitals and a model specifically for suspected UTIs in nursing homes used by McMaughan et al., where we revised the language for the setting and the users (Appendix A) [30,31].

##### Barriers to Implementation of the Communication Tool

The GP noted that the nursing home staff’s descriptions of what was wrong with the patient were sometimes vague:
“…We sometimes receive emails, where it says ‘the patient is ill, what should we do?’. And then we would like to, then we would like to go through some stuff, we need to have that specified.”

Therefore, in the Situation section of the ISBAR, the nursing home staff had to give an example of how the patients’ conditions had changed so that the GP could assess severity.

Previous opinion bias is when the diagnostic process is influenced by a previous assessment from another health professional, test result, or diagnosis [37]. When we interviewed the GP, he spontaneously said:
“…We are quickly influenced by the cause of enquiry… if an experienced nurse calls and tells us a lot of things from this box (points “new onset urinary tract symptoms”, red), well, then we don’t bother so much about this (points to “symptoms from other organs and other findings”, red)”

This suggests that previous opinion bias could be common in diagnosing UTIs. In addition, informants found that the Recommendation section in the ISBAR was too complicated. To simplify and avoid previous opinion bias, the section was changed to just pose the question “What do you think we should do?”

### 2.4. The Case-Based Education Session

#### 2.4.1. Content of the Case-Based Education

The initial content of the case-based education session was designed to minimise barriers impeding the implementation of the dialogue tool by addressing the knowledge gaps and biases found in the development process. The case-based education consisted of (a) a short, interactive lecture introducing the central concepts, followed by facilitated discussions, and (b) a case presentation and a case exercise. The facilitator presented the first case and modelled the use of the dialogue tool for the nursing home staff, who then applied the tool to the second case. After each case, the facilitator led a group discussion about the challenges involved in reworking previous notions of the diagnosis of UTIs through the dialogue tool.

In addition to the two cases, the topics discussed in the education session included the need for the intervention (i.e., the consequences of resistant bacteria and the communication pathway from nursing home resident to GP), possible knowledge gaps found in the tailoring process, and practical details about the trial.

The knowledge gaps we addressed were related to the definition of UTIs and asymptomatic bacteriuria. All informants in the tailoring process equated a positive urine test, smelly and unclear urine, with UTIs. A nurse in Phase 5 of the tailoring process put it this way:
“And so the urinary dipstick says nitrite and leucocytes and that’s what’s supposed to be there… This is what usually indicates an infection.”

Asymptomatic bacteriuria is frequent in nursing home residents; confusing UTIs with asymptomatic bacteriuria results in overtreatment [9]. Therefore, the definition of UTI from Loeb et al. was used as a starting point to divide symptoms of UTI into four groups: urinary tract symptoms, nonspecific symptoms, signs of bacteria in the urine, and severe symptoms [29]. Using this definition, the facilitator talked about asymptomatic bacteriuria, stressed that UTI is a clinical diagnosis with the presence of urinary tract symptoms, and that smelly and unclear urine could indicate the presence of bacteria, but that these signs have no diagnostic value [36].

The informants regarded a wide range of symptoms as nonspecific, e.g., falls, slurred speech, fatigue, and diarrhea. Evidence for a direct link between UTIs and nonspecific symptoms is ambiguous, but international guidelines propose that nonspecific symptoms should not be treated [9,10,38]. Boockvar et al. found that when nursing home residents displayed nonspecific symptoms, approximately 25% of them needed medical attention for various diseases, but the majority got better with no intervention [39]. In the development process, we observed that the nursing home staff would uncritically suspect a UTI when they observed a nonspecific symptom. In the focus group interview, a healthcare assistant said:
“Here it says significantly confused, and it is very typical for someone who has a UTI that she becomes confused and unsettled, like it says here, right. So that is very… I would say that this is straight by the book, right. But how people respond differs a lot.”

This perception is problematic because UTI is a diagnosis of exclusion. That means that the diagnosis should be reached by eliminating the alternatives [40]. Consequently, we developed an alternative approach to consider nonspecific symptoms, under the assumption that when nursing home residents display nonspecific symptoms, most suffer from something other than UTIs. The approach consists of four steps: First, the nursing home staff exclude as many somatic and nonsomatic causes as possible for nonspecific symptoms before suspecting a UTI. The principle is illustrated in Figure 3 and is called “The Reverse Triangle”. Second, the nursing home staff have to determine if the change is newly onset and significant. Third, the nursing home staff have to decide if they could wait and see, meanwhile initiating preventive measures. Fourth, if the staff decides to contact the physician, they have to provide specific examples to help the physician consider the severity of the nonspecific symptom.

#### 2.4.2. Adjustment to the Case-Based Education Session

When the case-based education was trialed in the pilot, the head nurse found that what helped them most in evaluating their residents with suspected UTI was “The Reverse Triangle”. Therefore, we illustrated “The Reverse Triangle” with a figure similar to Figure 3 in the teaching material to emphasise the concept in future education modules (See Appendix A).

The educational session was decreased from two hours to 75 min when the majority of the nursing home staff were invited in order to limit the resource burden on the nursing homes. In the pilot, owing to budget constraints at the nursing home, only the dayshift staff was able to participate, and therefore two educational sessions were sufficient. For the cluster-randomised trial, we estimated that three educational sessions were sufficient for each nursing home to include staff from the evening shift.

## 3. Discussion

This paper has described the process of developing a complex intervention for nursing home staff, as well as how the findings from the process have led to the refinement of the intervention. First, we found that the workflow in the nursing homes and the communication pathway from the bedside of the resident to the GP mean that the task of evaluating highly complex patients falls on the healthcare staff least trained for it. In addition, the information about the condition of the residents passes through many actors before reaching the GP. Our enhanced understanding of these issues led us to discard the point-of-care-test and focus solely on knowledge gaps and clinical reasoning among all nursing home staff. Second, the nursing home staff’s reports to the GP about the conditions of the residents often included insignificant deviations from the norm, and descriptions were often vague. To improve the GP’s chances of reaching a correct diagnosis, we adjusted the dialogue tool and the case-based education to emphasise that symptoms should be newly onset and significant deviations from the norm. Third, we determined that previous opinion bias could be a problem when the nursing home staff delivers the information to the physician. We tried to eliminate this by modifying the Recommendations section in the communication tool. Fourth, we addressed staff’s misunderstandings about nonspecific symptoms by emphasising that UTI is a diagnosis of exclusion and by introducing the concept of “The Reverse Triangle”. Finally, we were surprised to find that the nursing home staff had problems using the flowchart; we concluded that the preconceptions of our informants overruled the logic of the flowchart. Therefore, we made adjustments to the reflection tool to anchor the user to the flowchart path and inserted a discussion section to anchor the new definition of UTI at the nursing home. We also designed case-based education to address knowledge gaps and added discussions to improve the uptake of the new knowledge among nursing home staff.

### 3.1. Discussion of Findings

The problems that the informants experienced when using the flowchart may be explained by the dual-process theory of cognition [41]. According to this theory, human thinking and decision-making are divided into an intuitive system and an analytical system. The *intuitive system* is fast and effortless, drawing on readily available information and experience-based patterns. In clinical reasoning, it activates when a patient presentation appears familiar. The *analytical system*, however, is slow and requires deliberate reasoning and information gathering. In clinical reasoning, it activates if a patient presentation is perceived as complex and uncertain [42]. In the development process, the informants considered the presented case (see Text B1 in Appendix A) as a standard case of UTI and used their intuitive system for their diagnosis. In doing so, they made the systematic error (cognitive bias) of thinking that UTI was likely when it was not. When the interviewer pointed out to the informants that they reached the wrong conclusion, they recognised that using the reflection tool in the case would lead to the opposite conclusion, but they clearly expressed confusion and a discrepancy between their experience of UTI and the definition underlying the reflection tool.

Cognitive dissonance could explain this reaction. Cognitive dissonance is the experience of feeling mental discomfort in situations when attitudes, beliefs, or behaviours are in internal conflict. Humans will seek to reduce, eliminate, and avoid situations that can enhance discomfort [43]. In this instance, the nursing home staff’s existing knowledge and practices were misaligned with the definition of UTI underlying the reflection tool. When the two definitions differed, a conflict emerged between how the staff members usually assess a resident and how they should assess residents with suspected UTIs according to the tool. To eliminate the discomfort, informants may invent rationales that validate their own practice or invalidate the conclusion of the reflection tool. Consequently, lasting reductions in antibiotic prescriptions depend on acceptance of the new definition and a related practice of diagnosis. However, other factors might have influenced the informants’ reasoning and resulted in a wrong conclusion according to the flowchart. To ensure that the presentation of the case and the reflection tool were clear, the interviewer increasingly emphasised how to use the reflection tool, noting that only the observations from the case should be used to complete the flowchart.

In the development process, a pragmatic approach was often chosen. The pragmatic approach favours easy implementation and use of the dialogue tool, but may, in some cases, sustain the preconceptions of UTI, especially considering the discussion about cognitive dissonance. Specifically, we chose to include smelly and unclear urine, as well as the urinary dipstick result in the observation section of the tool, even though their diagnostic value is ambiguous. As many health professionals regard these as signs of UTI, we feared that omitting them could cause frustration among the staff, leading them to disregard the tool, and perhaps even cause conflict with the GP. Additionally, with the right input, the discussion section in the reflection tool could embed the new definition of UTIs; however, with the wrong input, it could enhance prevailing preconceptions.

### 3.2. Limitations and Strengths

The study has several limitations. The development process did not use an overall theoretical framework; hence, some aspects of implementation may have been overlooked. We involved the Senior Citizens’ Council Members as representatives of the nursing home residents because of the nature of the intervention. One of the Senior Citizens’ Council Members was a retired GP and one was a nurse; their background influenced their contributions, as their perspectives shifted between the patient perspective and their perspective as health professionals. Consequently, the patient perspective may be underrepresented in this study. Finally, the informants in our pilot were nurses, because holidays and illness prevented us from including other nursing home staff members.

The study also has several strengths. The development process engaged representatives from all stakeholder groups, as is recommended when developing complex interventions [26]. The development process also identified potential barriers in the setting and used this knowledge to improve the intervention. Specifically, the process changed the intervention from being primarily a decision aid to being more of a framework for clinical reflection. Previously, decision aids have been developed to curb the increasing overuse of antibiotics in nursing homes, but to our knowledge, a framework that depends upon more active reflection has not been reported before [22]. An active reflection framework conceptualises the inherent uncertainty in the UTI diagnosis and it supports the nursing home staff in contributing with their core competencies of intimate knowledge of the patient and preventive hygienic measures. Furthermore, increased attention to reflection and decreased focus on the diagnostic decision itself may deliver less biased information for the GP to arrive at a diagnosis. Finally, the case-based education follows principles that minimises bias and follows recommendations to increase clinical reasoning abilities in nursing [44,45].

Other interventions have had limited success with changing antibiotic-prescribing culture in nursing homes, and one study has confirmed the role of cognitive bias in antibiotic overprescription for UTIs [24,46]. This study suggests that the interplay between organizational structure and cognitive bias makes it difficult to change clinical practice in complex organisations. However, this intervention may be able to change the antibiotic-prescribing culture for UTIs in nursing homes, because it differs from previous interventions in four important ways: it used a rigorous tailoring process to develop the intervention, it targets all nursing home staff, and it increases focus on reflection rather than the diagnostic decision itself, leaving the GP to arrive at a diagnosis based on less biased information.

## 4. Materials and Methods

The research group was academically and professionally diverse, comprising varying approaches to and experiences with nursing homes and suspected UTIs. The group consisted of one general practitioner, one GP registrar, one medical doctor, one public health specialist, and one political scientist. The group had a close collaboration with a hygiene nurse working with several nursing homes.

### 4.1. The Original Understanding of the Field

The original understanding of diagnosis and treatment of UTIs in nursing homes was based on international literature and the clinical experience of the medical doctors in the group. We based our definition of a UTI in nursing home residents on the diagnostic criteria defined by Loeb et al. [29]. According to this definition, symptoms differ between nursing home residents with and without urinary catheters. For urinary catheter users, costovertebral tenderness, rigours, or delirium are symptoms of UTI. For non-urinary catheter users, only symptoms localised to the urinary tract indicate a UTI. This definition excludes nonspecific symptoms, such as confusion or aggression, for non-urinary catheter users as symptoms of UTI by default.

The nursing home staff observe the residents and then provide the clinical history to the physician [47]. Thus, sound knowledge of UTIs and good communication skills are central to providing sufficient and correct information for physicians to make an appropriate treatment decision. UTIs sometimes progress to severe illnesses. By including the measurement of CRP as a POC-test from the nursing home, we thought that physicians would be more likely to abstain from prescribing antibiotics, assuming that severe infections could be ruled out. Therefore, we originally hypothesised that prescription rates would decrease if guidance for obtaining and communicating a concise clinical history, together with the CRP measurement, was provided by the nursing home.

### 4.2. The Planned and Executed Developmental Stages

We developed and adapted the intervention through a multi-step iterative process with input from stakeholders.

Initially, we thought to include as stakeholders the nursing home staff, general practitioners and their staff, and even nursing home residents. Ultimately, we decided that as the interventions to be developed were aimed at health professionals, the nursing home residents should be excluded. Instead, we consulted Senior Citizens’ Council Members, who are elected officials in their municipality and serve as the link between the senior citizens and the city council, ensuring that the elderlies’ conditions, needs and wants are known and met [48]. This requires regular contact and dialogue with nursing home residents and their relatives. Thus, Senior Citizens’ Council Members provided a broad perspective on the acceptability of the intervention to nursing home residents and their relatives.

The intervention consisted of a dialogue tool and an educational component. We created an early version of the dialogue tool and then submitted it to a tailoring process. The early draft was developed by the primary investigator and, based on a thorough literature search, participatory observations at five nursing homes, semistructured interviews with staff from 4 nursing homes and 11 general practices (GPs and medical secretaries), and experiences from a quantitative survey in general practice about communication, diagnostics and treatment of UTIs in nursing home residents (unpublished).

We originally intended the tailoring process to consist of three focus groups with stakeholders [32]. However, after the first focus group interview, we realised that it was impossible to gather informants for the last two focus groups due to constraints on the informants’ time and resources. Therefore, the tailoring process included five separate phases containing (1) a focus group interview with nursing home staff, (2) a double interview with a general practitioner and medical secretary, (3) a double interview with two senior council members, and (4) and (5) two individual interviews with nursing home nurses. The sampling was purposive and assisted by the municipality of Gentofte, which was a collaborator in the project.

Finally, we conducted a nonrandomised pilot study, as defined in Eldridge et al. [49]. In the pilot study, the intervention was tested for one month at a nursing home with 60 resident beds. Revisions of the intervention were based on weekly conversations with the head nurse during the pilot and the evaluation of the pilot, consisting of two single interviews with nurses at the nursing home. The educational component of the intervention was developed during the tailoring process, using the information from the entire development process to inform the covered topics. For an overview of the developmental stages, see Table 2.

After each interview, members from the research group discussed the primary findings and how the intervention might be adapted to make the best use of those findings. The new draft of the dialogue tool was presented in the subsequent interview. In Phases 1–4 of the tailoring process, the dialogue tool was a rough sketch, but in Phase 5 and the pilot, it was redesigned using an online graphic design program (Appendix A).

### 4.3. Interviews during the Tailoring Process and the Pilot

In preparation for the interviews, we sent the latest edition of the dialogue tool by email to the informants. All interview guides were semistructured and contained the same themes for all stakeholders. These were opening statements, the dialogue tool, and advice on the implementation process. During opening statements, the informants were encouraged to discuss their immediate reaction to the dialogue tool and their concerns and experiences dealing with UTIs in nursing home residents. We then introduced the aim of the intervention and the way the dialogue tool was structured. For nursing home staff, the interviewer introduced a case (see Text B1 in Appendix A) so the staff could try the dialogue tool. In the individual interviews with nursing home nurses, we employed the “Think Aloud” method to gain a sense of the circumstances in which the use of the tool was suboptimal. We used the approach described in Boren and Ramsey [50]: The interviewer introduces the informant to the dialogue tool, the Think Aloud method and the case; the informants verbalise how they used the dialogue tool with the case, while the interviewer provides the acknowledgement tokens (e.g., “ok” and “mm”) to sustain the verbal report as undirected and as undisturbed as possible. The other stakeholders discussed concerns about the tool in more general terms from the perspectives of general practice and the patient. In the pilot study, we also discussed other aspects of the intervention, i.e., the educational component.

In interviews with more than one person, the research team sought consensus in discussions. A moderator and a comoderator facilitated the focus group interview. The moderator was responsible for the overall interview process, and the comoderator supported the moderator and was responsible for clinical content. A research assistant noted the group dynamics and the order of speech. One member of the research group conducted all subsequent interviews. Interviews were audio-recorded. Interviews with nursing home staff took place at the nursing homes, the interview with the GP and the medical secretary was held at the practice, and Senior Citizens’ Council Members were interviewed at the research groups’ facilities. All participants provided informed consent. Generated data were anonymised and kept confidential.

### 4.4. Ethical Approval

Because the study is not a health science project, as defined in the Danish Committee Act §2, the Research Ethics Committee of the Capital Region of Denmark waived the need for full ethical approval (Journal no: 17013412). The study was reported to the Danish Data Protection Agency.

## 5. Conclusions

This study has described how we confirmed and expanded our notion of the challenges faced by healthcare professionals when handling suspected UTIs in nursing home residents. We developed an intervention to improve upon the problems created by limited knowledge of UTIs among nursing home staff and the intricate communication pathway between nursing homes and general practice. The intervention underwent a tailoring process and a pilot to reduce barriers for implementation. Our main assessments were that GPs were often contacted about conditions persistent to the residents; previous opinion bias was present at the clinical handover to the GP; the predominant explanation of nonspecific symptoms was UTI; the intervention could be obstructed by what we perceived to be cognitive bias and cognitive dissonance. The specific changes prompted by these findings have been reported in this article to promote the transparency of the development process. The final intervention contains a dialogue tool comprised of a reflection and a communication component and a case-based educational session to address knowledge gaps and introduce the dialogue tool. Our intervention differs from previous ones in this area in four distinct ways: it used a rigorous tailoring process to develop the intervention; it targets all nursing home staff; it considers more causes of symptoms than just infectious disease; it focuses on reflection and less on decision-making. Overall, the study highlights a paradox created by the communication pathway from the bedside of the resident to the GP, namely, that the task of evaluating highly complex patients falls to the healthcare staff least trained for it. Furthermore, it suggests that lasting change in prescribing behaviour first requires changing nursing home staff’s beliefs about UTIs and how suspicions of UTI should be managed. Further research should explore the role of cognitive bias in relation to other health professions involved in the diagnostic process and different diseases managed in the nursing home setting.

## Figures and Tables

**Figure 1 antibiotics-09-00360-f001:**
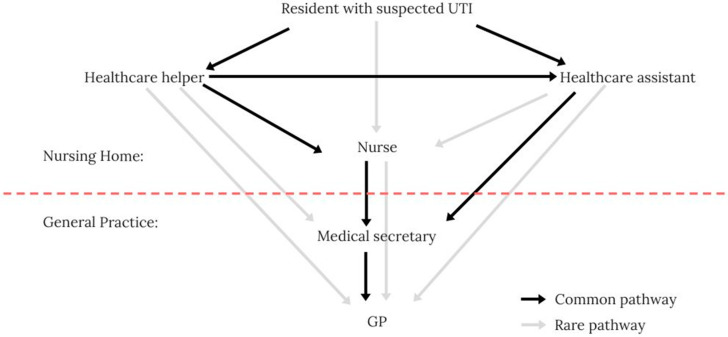
The typical telephone communication pathway between the nursing home and general practice.

**Figure 2 antibiotics-09-00360-f002:**
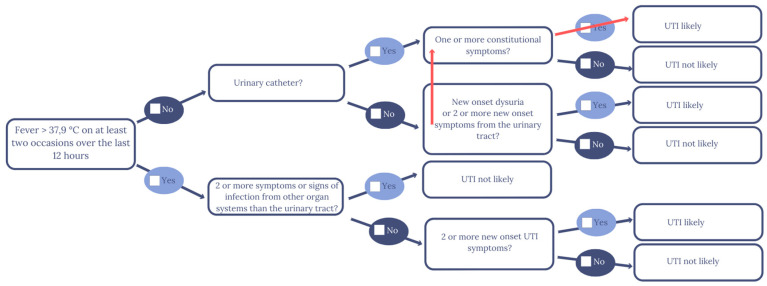
The healthcare assistant’s and nurse’s violation of the flowchart path in Phases 1 and 5 of the tailoring process.

**Figure 3 antibiotics-09-00360-f003:**
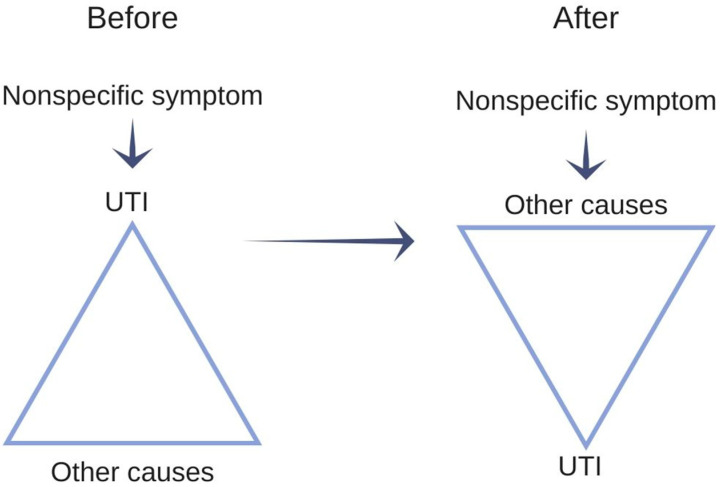
Illustration of the reverse triangle, similar to the one used in the case-based education material.

**Table 1 antibiotics-09-00360-t001:** Components of the original idea and the final intervention.

Components	Original Idea	Final Intervention
Diagnostic component	Decision aid by Loeb et al. [29]	Reflection (observations of signs and symptoms, flowchart and discussion)
Communication component	Communication tool by McMaughan et al. and Ydemann [30,31]	ISBAR (Identification, Situation, Background, Assessment, and Recommendation)
POC-test	CRP test	Discarded
Educational component	(1) Educational session to introduce decision aid and communication tool to all nursing home staff (2) Educational session to introduce CRP testing to selected staff	Case-based education to introduce dialogue tool and bridge knowledge gaps

**Table 2 antibiotics-09-00360-t002:** Developmental stages from the original idea to the final intervention of the dialogue tool.

Developmental Stage	Initial Draft	Tailoring					Nonrandomised Pilot
Phases	-	Phase 1	Phase 2	Phase 3	Phase 4	Phase 5	-
Date	April 2017–May 2018	June 2018	June 2018	June 2018	July 2018	July 2018	September 2018
Method(s)	Literature search Participatory observations Interviews Survey	Focus group interview	Double interview	Double interview	Single interview	Single interview	Two single interviews Four short telephone interviews during the pilot
Perspective	All	Nursing home	General practice	Patients and relatives	Nursing home	Nursing home	Nursing home
Informants background	Nursing home residents, all groups of nursing home staff, GPs, general practice staff	Three healthcare helpers, two healthcare assistants	One GP, one medical secretary	Two Senior Citizens’ Council Members	One nurse	One nurse	One head nurse, one nurse

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
