# Peer review of "Development of a Tailored, Complex Intervention for Clinical Reflection and Communication about Suspected Urinary Tract Infections in Nursing Home Residents"

_antibiotics, 2020, doi:10.3390/antibiotics9060360_

Round 1
Reviewer 1 Report
1. lines 109-110: "dipstick test" Indicate what this test does.
"ordering a urine culture" Indicate who can order this test.
2. line 120: "POC-test" Define this.
3. Table 1: Define "CRP test".
4. Line 135: "Loeb et al." is referred to, but the reference is not cited.
5. References are out of order. References 1 - 24 are in order, but the next cited reference is 30.
6. I could not find reference 48 cited in the text.
Author Response
Thank you for drawing our attention to these issues. Points 1-6 have been corrected - hopefully making the manuscript more readable in the process.
- lines 109-110: "dipstick test" Indicate what this test does. "ordering a urine culture" Indicate who can order this test.
Thank you for drawing our attention to this sentence. The sentence was not accurately conveying the intended meaning and we have rephrased.
The sentence was changed to (lines 110-112): “This system weighs objective, measurable results of urinary tests higher than subjective assessments of signs and symptoms when making the diagnosis.”
- line 120: "POC-test" Define this.
The POC-test is now defined in line 118 as ”…a C-reactive protein (CRP) point-of-care-test (POC-test).”
- Table 1: Define "CRP test".
”CRP test” should of course be defined before it is referenced in table 1. We moved definition from the methods section to the Results section (lines 118-119).
- Line 135: "Loeb et al." is referred to, but the reference is not cited.
The reference is now cited appropriately.
- References are out of order. References 1 - 24 are in order, but the next cited reference is 30.
We have checked the reference list thoroughly and it should now be in order.
- I could not find reference 48 cited in the text.
This error has been corrected.
Reviewer 2 Report
The manuscript entitled „Development of a Tailored, Complex Intervention for Professional Reflection and Clinical Communication about Suspected Urinary Tract Infection in Nursing Home Residents” by Arnold et al. describes the development of an intervention to decreased antibiotic consumption in the elderly for UTIs. The paper is definetly an important and significant contribution to the field.
Here are my recommendations for the paper:
The paper is generally well-written, however, the paper is very wordy in some parts. The reviewer recommends the involvement of a professional copy-Editor to shorten the paper and to correct the minor grammar issues.
Spaces are needed between reference numbers and the last words of the sentence.
Please use the passive voice in the text instead of active voice in the MS, especially when talking about aims of the study.
„Antibiotics are indispensable drugs, but the use causes development of resistant bacteria. Therefore, limiting unnecessary use is a public health priority.”
Please discuss this further, using some of these references:
Antibiotics 2020, 9(6), 310; https://doi.org/10.3390/antibiotics9060310
Molecules 2019, 24(5), 892; https://doi.org/10.3390/molecules24050892
Antibiotics 2020, 9(2), 41; https://doi.org/10.3390/antibiotics9020041
„Healthcare helpers and assistants have shorter education than nurses, and the education emphasises recognition of changes in the residents rather than knowledge pathology.” Please rephrase this sentence for more clarity.
Health professionals often consider nonspecific behavioral symptoms as an indication of UTI, but guidelines recommend that nonspecific symptoms in the elderly should not be treated with antibiotics.
Please include the following reference:
https://www.ncbi.nlm.nih.gov/pmc/articles/PMC6016416/
Please define and describe what/who constitutes as a „healthcare helper” The quality of Figure 2. should be improved.
Appendix should be capitalized in the text.
1
The Supplementary materials should be removed from the manuscript.
The authors should include the streghts and weaknesses of the paper in a separate section. The authors should elaborate more in the conclusion section.
2
Author Response
The manuscript entitled „Development of a Tailored, Complex Intervention for Professional Reflection and Clinical Communication about Suspected Urinary Tract Infection in Nursing Home Residents” by Arnold et al. describes the development of an intervention to decreased antibiotic consumption in the elderly for UTIs. The paper is definetly an important and significant contribution to the field.
Thank you for the kind words.
Here are my recommendations for the paper:
The paper is generally well-written, however, the paper is very wordy in some parts. The reviewer recommends the involvement of a professional copy-Editor to shorten the paper and to correct the minor grammar issues.
The paper has been edited and a native English speaker has corrected language and grammar issues.
Spaces are needed between reference numbers and the last words of the sentence.
This error has been corrected.
Please use the passive voice in the text instead of active voice in the MS, especially when talking about aims of the study.
We have changed to the passive voice several places. The objective and aim in particular was changed to (lines 79-82): “This paper will describe the development process of a tailored, complex intervention for a cluster randomised trial and the assumptions behind the intervention. The tailoring process was aimed at adjusting the intervention to the nursing home setting by identifying and addressing barriers to implementation.”
„Antibiotics are indispensable drugs, but the use causes development of resistant bacteria. Therefore, limiting unnecessary use is a public health priority.”
Please discuss this further, using some of these references:
Antibiotics 2020, 9(6), 310; https://doi.org/10.3390/antibiotics9060310
Molecules 2019, 24(5), 892; https://doi.org/10.3390/molecules24050892
Antibiotics 2020, 9(2), 41; https://doi.org/10.3390/antibiotics9020041
Thank you for this useful suggestion.
The last of the sentences mentioned above has been changed to (lines 42-44): “Therefore, preserving the effect of antibiotics by limiting unnecessary use is a public health priority and antibiotic stewardship is one way of achieving this goal.” “Molecules 2019, 24(5), 892; https://doi.org/10.3390/molecules24050892” have been referenced.
We felt that the reference “Antibiotics 2020, 9(6), 310; https://doi.org/10.3390/antibiotics9060310” would be a good fit in the last paragraph before the objective and aim to argue that (lines 74-75): “Studies have shown that uptake of antibiotic stewardship can be inadequate, perhaps because attention to implementation barriers is neglected.” We hope you agree.
„Healthcare helpers and assistants have shorter education than nurses, and the education emphasises recognition of changes in the residents rather than knowledge pathology.” Please rephrase this sentence for more clarity.
We agree that the sentence lacks clarity and have rephrased as (lines 60-62): ”While learning about disease and clinical reasoning is a central part of nursing education, it is virtually absent in the education of a healthcare helper and assistant.”
Health professionals often consider nonspecific behavioural symptoms as an indication of UTI, but guidelines recommend that nonspecific symptoms in the elderly should not be treated with antibiotics.
Please include the following reference:
https://www.ncbi.nlm.nih.gov/pmc/articles/PMC6016416/
Thank you for bringing this article to our attention. It has been included as a reference to the suggested sentence.
Please define and describe what/who constitutes as a „healthcare helper”
We have specified what/who constitutes a healthcare helper by clarifying the educational differences between a healthcare helper and a healthcare assistant. Lines 58-60 now reads: “Healthcare helpers undergo 19 months of schooling after their compulsory education, which can be shortened with additional basic schooling. Healthcare helpers become healthcare assistants by continuing their education an additional 20 months.”
The quality of Figure 2. should be improved.
We have redone Figure 2 and improved the solution.
Appendix should be capitalized in the text.
We have capitalized Appendix in the text.
The Supplementary materials should be removed from the manuscript.
We have removed the supplementary materials from the manuscript.
The authors should include the streghts and weaknesses of the paper in a separate section.
Limitations and strengths now have a separate section in the discussion.
The authors should elaborate more in the conclusion section.
We recognise that the conclusion has mainly addressed the findings on cognitive bias and dissonance. We have rewritten the entire conclusion and it now reads (lines 516-537): “This study has described how we confirmed and expanded our notion of the challenges faced by healthcare professionals when handling suspected UTIs in nursing home residents. We developed an intervention to improve upon the problems created from limited knowledge of UTI among the nursing home staff and the intricate communication pathway between nursing homes and general practice. The intervention underwent a tailoring process and a pilot to reduce barriers for implementation. Our main assessments were that: GPs were often contacted about conditions persistent to the residents; previous opinion bias was present at the clinical handover to the GP; the predominant explanation of nonspecific symptoms was UTI; and the intervention could be obstructed by what we perceived to be cognitive bias and cognitive dissonance. The specific changes prompted by these findings have been reported in this article to promote transparency of the development process. The final intervention contains a dialogue tool comprising a reflection and a communication component and a case-based educational session to address knowledge gaps and introduce the dialogue tool. Our intervention differs from previous ones in this area in four distinct ways: it uses a rigorous tailoring process to develop the intervention; it targets all nursing home staff; it considers more causes of symptoms than just infectious disease; and it focuses on reflection and less on decision. Overall, the study highlights a paradox created by the communication pathway from the bedside of the resident to the GP, namely that the task of evaluating highly complex patients falls to the healthcare staff least trained for it. Furthermore, it suggests that lasting change in prescribing behaviour first requires changing nursing home staff’s beliefs about UTIs and how suspicions of UTI should be managed. Further research should explore the role of cognitive bias in relation to other health professions involved in the diagnostic process and different diseases managed in the nursing home setting.”
Reviewer 3 Report
This paper describes the development process of a tailored, complex intervention for a cluster randomised trial and the assumptions behind the intervention. The authors reported that sustainable change in antibiotic prescribing behavior in nursing homes requires a change in nursing home staff’s beliefs about the management of urinary tract infections. In my view, this is an interesting article. However, it should be revised carefully regarding both the contents and language before further consideration.
Here are some remarks to this manuscript:
1. The technical writing is good but the English writing needs to be improved as there are many awkward phrasings and grammatical errors.
2. I think there is a misprint with the title. Please check it.
3. Keywords: “interprofessional relationship; medical education; organizational change” are not eligible keywords. I suggest deleting them.
4. As far as I know, last month a similar paper has been reported by the authors (JMIR Res Protoc, 2020 May; 9(5): e17710. doi: 10.2196/17710). So please justify what is the significance or advancement of this work in comparison with your previous publication?
5. Line 92: The abbreviation GP should be defined at its first mention in the manuscript.
6. I strongly recommend the authors refine the section of 2.3.1.2 because it is too interminable.
Author Response
This paper describes the development process of a tailored, complex intervention for a cluster randomised trial and the assumptions behind the intervention. The authors reported that sustainable change in antibiotic prescribing behavior in nursing homes requires a change in nursing home staff’s beliefs about the management of urinary tract infections. In my view, this is an interesting article. However, it should be revised carefully regarding both the contents and language before further consideration.
Thank you for the kind words.
- The technical writing is good but the English writing needs to be improved as there are many awkward phrasings and grammatical errors.
We have carefully revised the content of the manuscript and a native English speaker has revised language and grammar.
- I think there is a misprint with the title. Please check it.
Thank you for pointing this out. We have removed the word “Title” in the title.
- Keywords: “interprofessional relationship; medical education; organizational change” are not eligible keywords. I suggest deleting them.
We agree that these keywords are not eligible and have deleted them.
- As far as I know, last month a similar paper has been reported by the authors (JMIR Res Protoc, 2020 May; 9(5): e17710. doi: 10.2196/17710). So please justify what is the significance or advancement of this work in comparison with your previous publication?
It is correct that we published the protocol of the cluster randomised study measuring the effect of the intervention last month. The protocol describes the final intervention used in the cRCT, whereas this paper addresses the development and refinement of the intervention on the basis of the identified barriers to implementation we found through the tailoring process and the pilot. In addition, a Cochrane review from 2015 about tailored interventions concluded that tailoring is “not described in detail in published studies”[1]. Therefore, we argue that the present study is a significant contribution to the literature because of the methodological description and the findings relevant to other antibiotic stewardship interventions.
- Baker, R.; Gillies, C.; Ej, S.; Cheater, F.; Flottorp, S.; Robertson, N.; Wensing, M.; Fiander, M.; Mp, E.; J, V.L.; et al. Tailored interventions to address determinants of practice. Cochrane Database Syst. Rev. 2015, doi:10.1002/14651858.CD005470.pub3.www.cochranelibrary.com.
- Line 92: The abbreviation GP should be defined at its first mention in the manuscript.
Thank you for bringing this to our attention. We have defined the abbreviation GP at its first mention in the manuscript.
- I strongly recommend the authors refine the section of 2.3.1.2 because it is too interminable.
We have revised and shortened section 2.3.1.2 substantially. Please see the track changes document for the specific changes.